# ε-Rotation Invariant Euclidean Spheres Packing in Slicer3D

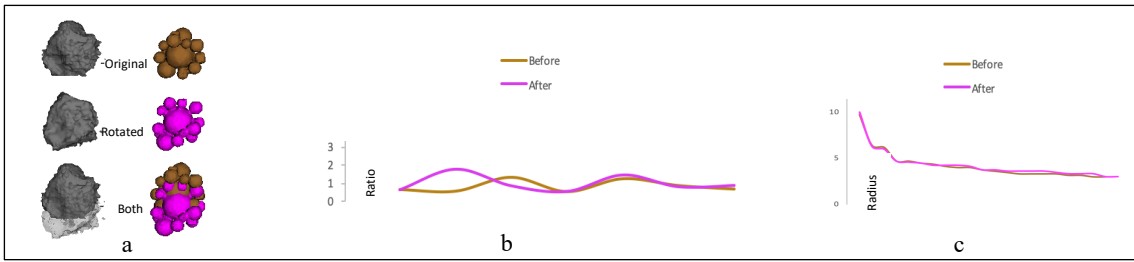

Figure 1: a: Original and rotated 3D volume visualization with its spheres packing. b: Difference in the distance ratios between sphere's centers before and after rotation. c: Difference in the sphere's radiuses before and after rotation.

## ABSTRACT:

Sometimes SRS (Stereotactic Radio Surgery) requires using sphere packing on a Region of Interest (ROI) such as cancer to determine a treatment plan. We have developed a sphere packing algorithm which packs non-intersecting spheres inside the ROI. The region of interest in our case are those voxels which are identified as cancer tissues. In this paper, we analyze the rotational invariant properties of our sphere-packing algorithm which is based on distance transformations. ε-Rotation invariant means the ability to arbitrary rotate the 3D ROI while keeping the volume properties remaining (almost) same within some limit of ε. The applied rotations produce spherical packing which remains highly correlated as we analyze the geometrically properties of sphere packing before and after the rotation of the volume data for the ROI. Our novel sphere packing algorithm has high degree of rotation invariance within the range of $\pm$ ε. Our method used a shape descriptor derived from the values of the disjoint set of spheres form the distance-based sphere packing algorithm to extract the invariant descriptor from the ROI. We demonstrated by implementing these ideas using Slicer3D platform available for our research. The data is based on sing MRI Stereotactic images. We presented several performance results on different benchmarks data of over 30 patients in Slicer3D platform.

## KEYWORDS:

Rotation Invariant, Slicer3D, Sphere Packing, Distance Transformation, Stereotactic.

## INTRODUCTION:

In several applications such as inspection of tumor or interacting with portion of a 3D volume data, the ROI could be rotated at arbitrary angles. If a sphere packing algorithm is used before and after such rotation, then rotational invariance suggests that there might be high correlation between spheres found by our sphere packing algorithm before and after the rotation. Defining correspondences between the original and rotated ROIs is an important task that could be solved by spheres' descriptors. If these descriptors are highly correlated, then we can anticipate that the ROIs might be similar as well. Li et al. (Li & Simske, 2002) stated that translation and scaling are easy compared to rotation. Rotation of a 3D volume data or 3D image involves simultaneous manipulation of three coordinates to maintain invariance. In the case of sphere packing, as we capture the ROI with non-intersecting spheres, the rotation invariance means that set of spheres will remain identical in size although their placement is expected to change under an arbitrary rotation. There are three major techniques to prove the rotation invariance: landmarking, rotation invariant features/shape extraction descriptor, and brute force rotation alignment. *The landmarking* is normally carried out by following two methods, domain specific landmarking and generic landmarking (Szeptycki, Ardabilian, & Chen, 2009). The domain specific landmarking accepts some fixed point in the image and does rotation with respect to that about an arbitrary axis. The generic landmarking method on the other hand, finds the major axes of the 3D/2D image and that can rotate the volume or image as a whole in carrying out the rotation. Because the size of the volume data can be typically large based on the size of the data, both these approaches require that large memory storage is available as the complete voxel information is required, and usually is time consuming. *The brute force alignment* method divides/degrades the object into large number of smaller parts and works with them for rotation. This method is time consuming, complex and complicated because *parts* have to be organized. The developed code for a particular shape in this method may only

apply to the data in hand and may not be generalizable. Finally, *Invariant feature/shape descriptor* involves identification of certain invariant features (measurable quantities) that remains unaltered under rotations of the 3D image or volume data. The invariant features are indexed with a feature vector also known as shape signatures. Then, the optimal rotation can be defined by measuring model's similarities in terms of the *distance* such that the rotation invariant property would mean that these distance measures are as close to each other with certain limit before and after the rotation. There are literally many of rotation invariant features that been used in the past, including ratio of perimeter to area, fractal measures, circularity, min/max/mean curvature, and shape histograms, etc. Lin et al. (Lin, Khade, & Li, 2012) and Yankov et al. (Yankov, Keogh, Wei, Xi, & Hodges, 2008) use time series representation as a feature vector to match the 3D shapes to prove the rotation invariance. Based on our research, most of the studies have been used spherical harmonic method to map the features of objects into a unit sphere to prove the invariance under rotation (Kazhdan, Funkhouser, & Rusinkiewicz, 2003; Nina-Paravecino & Manian, 2010; Vranic, 2003). The spherical harmonic method does not always give accurate results to distinguish between models since the internal parts of the 3D shapes may not fit in same sphere. Other researchers combined the spherical harmonic with spatial geometric moments (El Mallahi, Zouhri, El-Mekkaoui, & Qjidaa, 2017; Kakarala & Mao, 2010). The most common graph method used is skeletons. The skeletons are based on medial axis. The medial axis of the 3D objects has been used as a shape descriptor in a number of researches (Iyer, Kalyanaraman, Lou, Jayanti, & Ramani, 2003; Iyer, Jayanti, Lou, Kalyanaraman, & Ramani, 2004; Liu, 2009; Lou et al., 2003; S'anchez-Cruz & Bribiesca, 2003; Sundar, Silver, Gagvani, & Dickinson, 2003). However, this method is sensitive to noise and has a heavy computationally cost.

In this paper, we considered the set of spheres as shape-descriptors and analyzed the sphere packing before and after the rotations and looked for the similarity measure. We aimed to show that set of spheres are invariant such that even if we rotate the image, the size of the spheres and center's distances are highly correlated. We used our sphere packing algorithm to pack non-intersecting spheres into the ROIs before and after rotations. As mentioned earlier, those spheres could provide invariant shape descriptor. After rotation the voxels will be populated with the new voxel orientation. Our shape descriptor provides a novel featureless method that doesn't depend on any specific feature or texture, instead is related to sphere packing generated by our sphere packing algorithm. Our method characterizes the 3D object similarity by

the shape geometries of the sphere packing, and sphere's correspondence with one another and their spatial relationships. In this paper, we show that our previous work for sphere packing (Anonymous, 2019) can be used to show the invariance under rotation since our algorithm can describe volumetric shapes more succinctly than voxel representation. In this work, the spheres packing together with the radiuses and centers functions provided an shape descriptor, a novel approach for characterization and compression of shape information for 3D volume and voxel data.

## SPHERE PACKING DESCRIPTOR

In Stereotactic radio surgery, tumors are irradiated by beams of high-energy waves. It is a challenge during cancer treatment planning to provide minimal damage to healthy tissue around the tumors that get exposed to the radiation and still radiate cancerous cells. Our goal using sphere packing is to arrange beams on "spheres" in a way that hit the unhealthy tissue and leave the healthy tissue intact. A key geometric problem in Stereotactic radio surgery planning is to fill a 3D irregular-tumor shape (ROI) with disjointed spheres. We use the spheres packing to represent the 3D object, so this method of representation is called *region-based* descriptor since it is based on regions. In one of our work, Sphere Packing algorithm is used based on the maximum Euclidean distance has been studied and implemented in Slicer3D using medical imaging (Anonymous, 2019). Sphere packing problem is heuristically solved by using Euclidean maximum distance. The solution is to find a set of non-intersecting spheres that used greedy method and can be called largest sphere first. Each sphere is characterized by its radius and center. The size of the regions can be controlled depending on the treatment planning required size. Also, in our implementation, the density of the volume coverage can be customized such as we did in (Anonymous, 2019), we used 50%-90% of the density. This means 50% to 90% (or theoretically any amount up to 100%) of the ROI is covered disjoint spheres which our algorithm finds. Of course, more the coverage, more time is taken by the algorithm to find all the spheres satisfying the user selected criteria. Generally for all patients, 50% coverage takes up to 25 minutes, and 90% takes minimum of 7 hours and maximum of 72 hours.

Our algorithm for the sphere packing is defined as a set of n unequal spheres and object P of a bounding box B. Each sphere $i \in n = \{1, 2, \ldots, n\}$ is characterized by its radius $r_i$ and center $c_i$. The goal of this algorithm is to pack sets of disjoint spheres inside the ROI providing certain coverage. Our strategy is as follows: A uniform grid (voxelization) is used to calculate the maximum distance of each voxel to the

3D object boundary. Then, use the maximum distance to be the radius of the first sphere and the location to be the sphere center. Iteratively, we extract new spheres each time and recalculate the distances based on the following constraints: spheres must not intersect with other spheres must completely locate inside the volume, and the volume covered by spheres is maximized using greedy strategy by subtracting the volume of the largest sphere for every iteration where largest sphere is found using a distance transformation. In our technique, sphere placements are no longer on the skeleton line. Instead, the spheres are placed wherever the maximum distance value occurred inside the ROI during that iteration (Fig. 2). We applied our maximum distance sphere packing strategy algorithm successfully on many MRIs using the Slicer3D platform; a new module in Slicer3D to be used for different shape approximation purposes.

The spheres centers of the 3D object represent a spatial template as a graph. The graph is a representation of the intersection of the sphere's centers that represent vertices of the graph of all maximum distances contained inside the 3D object, and edges connected each two consecutive generated spheres (Fig. 3). Ordering of the spheres is important for example: B, C, A will give different signature graph than A, B, C.

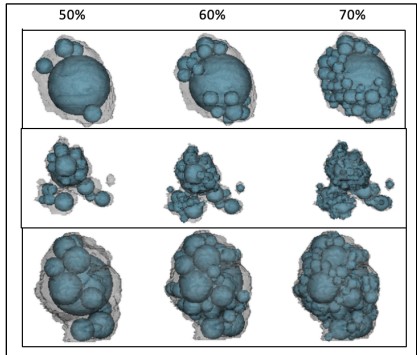

Figure 2: 3D results of our algorithm for sphere packing in Slicer3D with 50%, 60%, 70% of packing density (gray is tumor, blue is sphere) (Anonymous, 2019).

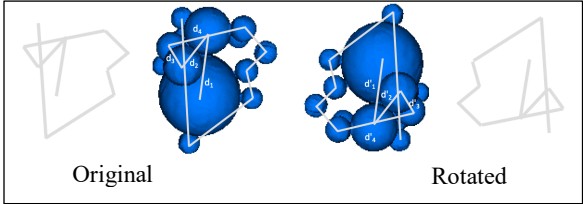

Figure 3: Spatial template graph of the original and rotated volumes generated from the intersection of the spheres' centers.

## EPSILON ROTATION INVARIANT:

We introduce a measure called epsilon-rotation invariant. Such geometric accuracy of MRI is practical especially when it used for planning radio surgery. Testing different angles of the image for beam planning is needed. Rotating the 3D volume must give the similar arrangement of sphere packing. We captured inner distances between two consecutive spheres' *centers* of our shape descriptors as an approximation to compute the difference between the two 3D shape descriptors, before and after the rotation. This graph distances representation is useful to abstract a geometric meaning of the 3D shape and to characterize the connectivity information. From Figure 2, assuming we rotated a 3D tumor, apart from how close $d_1$ is to $d`_1$ and $d_2$ is to $d`_2$ etc., we also look at the inner spheres' centers distances between center of the original spheres compared with the corresponding distances on the rotated volume by finding the ratio as follow:

$$\text{Distance ratio} = \begin{cases} d_1/d_2 = d`_1/d`_2 & \text{invariance is met} \\ d_1/d_2 \neq d`_1/d`_2 & \text{invariance not met} \end{cases}$$

The inner distances between the spheres capture the distances before and after the rotation of 3D object, and find the sphere packing descriptors. In other words, we find how similar the spherical coverage is before and after, and intuitively compare that to the graph inside the spheres. Although we did not implement the orientation of such inter-distances, we expect that to be closely related for better results for our distance transformation based shape descriptors. Intuitive idea is that apart from radius being equal, the relationship between the centers should also be similar between one sphere to another. Our algorithm descriptor map entries correspond to the Euclidean distance between spheres' centers and these values are arranged in a manner that preserve the relative position of each sphere.

## IMPLEMENTATION:

We implemented our method in Slicer3D (Fig. 4). The Slicer3D (Kikinis, Pieper, & Vosburgh, 2014), is an open source medical visualization tool. The 3D slicer builds on top of different libraries such as VTK, ITK, CMake, NA-MIC, Qt and Python (Anonymous, 2018). Also, it contains more than a hundred modules written in C++ or Python to provide researchers many common tools and rich implementations to achieve and implement their goals. The visualization toolkit (VTK) framework is an open source with C++ libraries that contains many filters for data representation/visualization. We developed our Slicer3D Python module for sphere packing to work with the VTK for volume rotation. 3D arbitrary

rotations are introduced for medical images as an extension of our previous work carried out for sphere packing (Anonymous, 2019). We used *VtkTransform* to apply rotation via 4x4 matrix multiplications. Our algorithm rotates images any number of degrees around x, y, and z axes. Any arbitrary rotation can be described by specifying the coordinates of the object in 3D space and rotation angels. Unlike 2D rotation, 3D rotation occurs along an arbitrary axis. Suppose the rotation angle is a, the rotations about three major axes uses well known formulae:

- Rotation along x= $\begin{bmatrix} 1 & 0 & 0 \\ 0 & \cos(a) & -\sin(a) \\ 0 & \sin(a) & \cos(a) \end{bmatrix}$

- Rotation along y= $\begin{bmatrix} \cos(a) & 0 & \sin(a) \\ 0 & 1 & 0 \\ -\sin(a) & 0 & \cos(a) \end{bmatrix}$

- Rotation along z= $\begin{bmatrix} \cos(a) & -\sin(a) & 0 \\ \sin(a) & \cos(a) & 0 \\ 0 & 0 & 1 \end{bmatrix}$

Slicer3D create a 3D scene file as Medical Reality Markup Language (MRML) and display images in physical space using patient coordinates system RAS (Right Anterior Superior), based on the information of the image spacing, origin, and direction. When applying rotation, we used the spheres packing information along with the origin and spacing. Thus, before we apply the rotation, we need to know the data of the volume:
- Position: the 3D coordinates of the object.
- Bound: the bound box of the object represented as (xmin, xmax, ymin, ymax, zmin, zmax).
- Origin: it is the position of the first voxel in the patient coordinate (0, 0, 0). It is the space origin, which is the center of all rotations
- Spacing: it is the voxels distances along each axis in the image.

Applying rotation using VtkTransform is done by following six phases as follow:
- **Phase 1**: Crate and add a transformation node. We first create a TransformNode using VtkMRMLTransformNode, then add that node to the MRML scene. This node contains the transform ID and can store any linear transformations of composite of multiple transformations.

```
def addTransform(self):
 transformNode = slicer.mrmlScene.AddNode
            (slicer.vtkMRMLTransformNode())
```

- **Phase 2**: Create a homogenous 4x4 transformation matrix. VtkTransform generates

4x4 matrix that initialized to the identity matrix transformation (all zeros with ones in the diagonal) to describe the linear transformation.

```
Rotation = vtk.vtkTransform()
```

- **Phase 3**: Set the parameters of rotation. Our algorithm rotates images any number of degrees around x, y, and z in z, x, y order.

```
if rx != 0:
    rotation.RotateX(rx)
if ry != 0:
    rotation.RotateY(ry)
if rz != 0:
    rotation.RotateZ(rz)
```

RotateX, RotateY, and RotateZ create the rotation matrix. Since VtkTransform rotate the object around the origin (0,0,0), the rotation algorithm performs the following steps to rotate the volume about its center. The volume is first translated to its center so that its centroids lie on the center of the image instead of the origin (0,0,0). The resulting volume is then rotated according to transformation chosen by the user (x, y, and z angles values). Then, translate the volume back to its original pose.

- **Phase 4:** Apply transformation.

```
tNode.ApplyTransformMatrix(rotation.GetMatrix())
```

Where tNode is the transform node. GetMatrix is used to return the current values to be used for the view manipulations such as rotate the current values in x, y, z angles. So, the current values (vtkMatrix 4x4) are multiplied by transformation matrix. Applying transformation is basically done by multiplying current node with the transform node and stored in simple linear transform:

```
VtkMRMLVolumeNode(tNode) * transformNode
```

- **Phase 5:** Concatenate multiple (nested) transformations and attach volume to transform node.

```
OutputVolume.SetAndObserveTransformNodeID(tNode.GetID())
```

- **Phase 6:** Harden transform.
  Describe applying transformations and save it as a transformed model. Invoking transform model and *harden* transform the volume to get the correct new orientation, which will be stored in the image header. Thus, *harden* transformation is

used for: changing orientation and generation of the output volume.

```
logicH = slicer.vtkSlicerTransformLogic()
logicH.hardenTransform(outputVolume)
```

Our heuristic is based on the greedy concept of generating largest spheres first, is $O(n*K)$ where n is the number of spheres found to satisfy the chosen coverage and K is per sphere iterative constant where K= number of voxels in the volume data set. The pseudo code for Euclidean sphere packing rotation algorithm is as follow:

```
1.  Input volume as nrrd file
2.  Calculate 3D coordinates position, origin,
    and spacing
3.  Add a transfer node: VtkMRMLTransformNode
4.  Create a homogenous 4x4 transformation
    matrix: VtkTransform
5.  Set the parameters of rotation: RotateX(),
    RotateY(), RotateZ()
6.  Apply transformation: ApplyTransform
7.  Compute the new spacing, origin
8.  Output (Rotated Model)
9.  SegmentedRegion= Bounded rotated Model
10. Distance= EuclideanDistance(SegmentedRegion)
11. Max= maximum distance
12. Sphere_raduis= max
13. Sphere_center= location(max)          // center
    inside bounding box
14. Sphere_isocenter= (Sphere_center + nrrd)  //center
    inside nrrd volume
15. Sphere_grid= (distance, radius, center)
16. SegmentedRegion= (1- Sphere_grid)
17. While(pixels not covered > pixels in desired
    coverage)do
         Repeat Steps 10-16
18. Draw MultiSpheres()
```

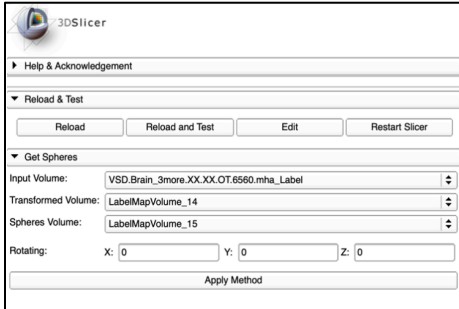

Figure 4: The GUI of our sphere packing rotation in Slicer3D.

## RESULTS:

Experimental results demonstrate the effectiveness and efficiency of the proposed method. In our experiments, we used thirty MRIs of segmented brain tumors from the BRATS dataset (Menze, Bjoern H and Jakab, 2015) separated on three datasets with ten patients on each (Fig. 5). The three datasets are manually revised and delineated by experts broad-certified neuroradiologists and radically different in size, shape and complexity. The tumor sizes range from 248318 to 12948 pixels.

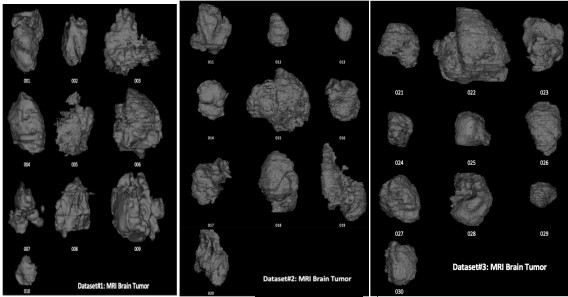

Figure 5: 3D MRI datasets of brain tumor.

The proposed method is applicable to different volume shape representations and different arbitrary rotations angles (Fig. 6). Our attempt for seeking a rotation invariant descriptor is to improve the matching similarity performance inspired by region's partition by non-intersecting spheres packing and based on maximum distance. We first divide the volume with spheres. Then, distances ratios and radii of each sphere are calculated to be compared with the correspondence on the rotated volume of distance ratio and radii values (Fig. 1). More similarity measurements are investigated such calculating the accuracy along with the Mean Absolute Error (MAE) of our algorithm. We developed an epsilon value measure for similarity based on our study. This allowed us to manage differences that result due to the fact that voxel sizes also change and are within epsilon ($\varepsilon$) of each other.

## Results under Epsilon ($\varepsilon$) -value criteria:

In our study, we observed interesting patterns looking at the radius of spheres, and they are being close enough before and after the rotation. The spheres radiuses and distance ratios are actually within epsilon ($\varepsilon$) value criteria. The Epsilon is the maximum distance in terms of voxel size and is always given within a small range of numbers. Thus, after analyzing our 30 patients MRIs, the epsilon values of the difference spheres' radius before and after rotation are under one unit of difference, specifically within the value of 0.8 mm. Therefore, any radiuses within $\pm$ 0.8 are meant to be acceptable and there is then a high probability that the 3D volumes are similar when there are no multiple spheres with the same sphere-radius. Since previous epsilon value is based on the 50% of the packing density, we tested our epsilon value under different packing densities such as 60%, 70% and 90%. We found that, our epsilon value is consistent under $\pm$ 0.8 (Fig. 7).

On the other hand, when we consider the ratio of distance between two consecutive spheres in both before and after sphere packing list, the epsilon value between the *distance ratios* of the original volume and the rotated volume is within ± 2.5. That means, the difference in distance ratio between any consecutive spheres has to be within ± 2.5. However, increasing the packing density strongly increase this value to ± 4 in 60%, ± 12 in 70%, and ± 32 in 80% of packing density (Fig. 8). This is as expected results (discussed in the next section) as we go deeper in the list of spheres.

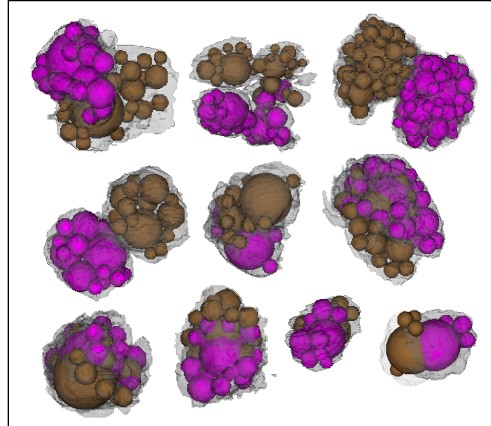

Figure 6: Rotational Euclidean sphere packing.

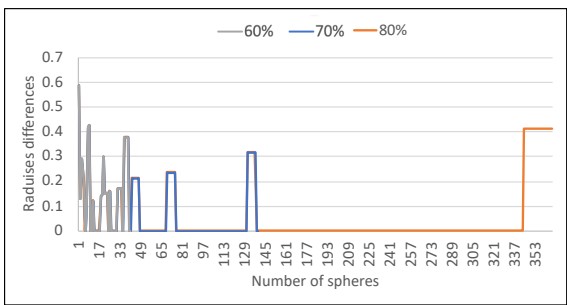

Figure 7: Increase the number of spheres within epsilon value with the increase of spheres packing density.

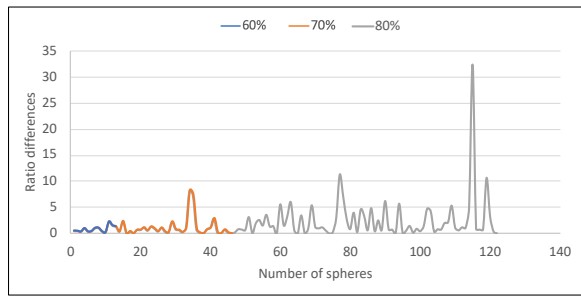

Figure 8: Increase the epsilon value with the increase of spheres packing density.

The accuracy of our technique to approximate the same radiuses/ratio values after rotation is calculated for each patient data, we divided each radius or ratio in the original volume by its corresponding radius/ratio after rotation to see how well our algorithm works. The overall average radiuses accuracy percentage is 96.86% (Fig. 9). On the other hand, differences between the ratio of distance before and after, the overall Ratio average accuracy of getting same ratios is 69.23% (Fig. 10). We noted that our results are driven by a good accuracy and further 3D-spatial improvements will fetch better results.

To increase the accuracy, we further tested the accuracy of some of our results by varying the volume dimension (voxel size). Our datasets patients' grid size varies so we increased the dimensions to different values to have bigger grid size with a greater number of voxels of smaller sizes. We find that the decrease in voxel size change the accuracy level. The more the voxel size decreased (grid became finer), the accuracy increased (Fig. 11). This is expected because smaller voxel sized provide more accuracy than the bigger size

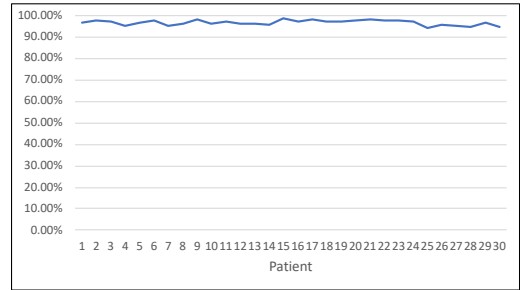

Figure 9: Radiuses accuracy.

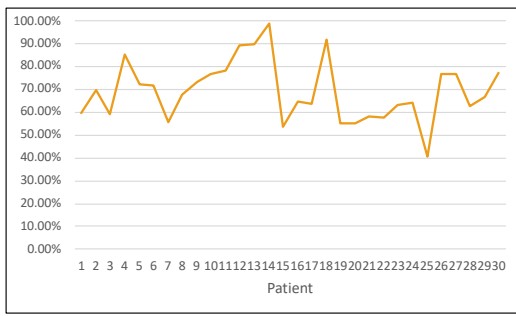

Figure 10: Distance ratio accuracy.

Still, we analyzed the data further. For each patient, we computed the absolute error between the original and the rotated radiuses/ratios. Absolute values were estimated across a range of different patients.

```
Absolute error = (Before_radius – After_raduis).
```

Then, the mean absolute error (MAE) was calculated for patients using the distribution of ratios and radiuses in 30 patients. The closer this value to the zero indicate

the great algorithm approximation to cover the targeted object. The overall MAE of our algorithm is 0.2. In medical applications (Irwig, 2007), we believe that this measure could be important because the absolute error represents the risk of developing recurrent disease because this value indicates the untreated cells/voxels. Being able to differentiate between patient with highest and lowest absolute risk of recurrence is an important task in order to diagnose the patient with the appropriate treatment. Therefore, the MAE play an important role to differentiate whom radiotherapy can yield to meaningful benefits.

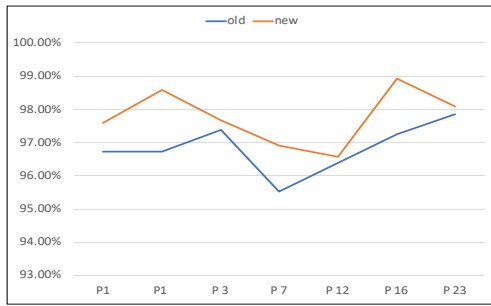
Figure 11: Increased radius accuracy.

## DISCUSSION:
The spheres radius works the best for our study for finding the similarity after rotation. Even though there are differences between the total number of calculated distances before and after rotation, our algorithm accuracy is reasonably high because it is able to calculate *almost* similar radiuses each time within epsilon. The consistency of spheres radius is because our algorithm at each iteration finds the maximum radius distance to pick first, so increasing the number of packed spheres to cover the required voxels based on the desired packing density doesn't affect the epsilon value. Changing the topology due to equal spheres is the main reason of the increase of the epsilon value of the distance ratios. The algorithm decision of choosing which sphere of the same size to place first, is the big issue here. Therefore, increasing the number of packed spheres will significantly increase the changes in topology which will results in increasing the epsilon value.

When the radiuses are equal, the descriptor graph before and after might change considerably based on which sphere our algorithm suggests. In this case, aggregate we will need to collect all those spheres which are equal and replace them by the average of the center of the sphere in the shape descriptor, and so then ($\varepsilon$) value will be similar in the shape description before and after the rotation. Thus, set of spheres whose radius are equal are replaced with one sphere. That is expected to reduce the epsilon ($\varepsilon$) value further. Moreover, the topology changing in our study affect

our accuracy results. We believe that eliminating equal spheres by using the enclosing sphere in our implementation will decrease the distance ratio results comparing the shape descriptors before and after the rotation of the ROI.

## CONCLUSION:
Our novel medical visualization techniques promise to improve the efficiency, diagnostic quality and the treatment. The field of 3D shape approximation and similarity have been a focus in the area of geometry in general for several hundred years now. Shape analysis for feature extraction is the key problem in the shape approximation and similarity issues. The best way for similarity matching is to identify certain shape signatures (prominent features in the image). These signatures are then compared between the transformed images through similarity assessment, distance computation or any other appropriate methods. This paper presented a method for defining a possible invariant shape descriptor from 3D-image or 3D-volume data to be used to match the objects from different rotations/viewpoints. Our method can be applied to a wide variety of data types such as 2D images and even polygonal meshes. Our heuristics is $\varepsilon$-invariant and has an impressive result of 96% invariant under rotations. The experimental results prove the effectiveness of our novel idea. The proposed system was fully software implemented in Slicer3D and has been tested on 30 patient's databases. For future works, we will apply other measures such as 3D-spatial sorting based on the spheres found, or identifying a minimal volume enclosing sphere surrounding all spheres of equal radius (as mentioned earlier) to improve epsilon ($\varepsilon$) value further. Moreover, as Slicer3D is experimental, not FDA approved, yet used worldwide, our plan is to upload our implementations under BSD license so that world-wide communities can try the system and provide more feedback using their 3D volume data and reporting $\varepsilon$-value for their data.

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
