# OpenReview forum: "epsilon-Rotation Invariant Euclidean Spheres Packing in Slicer3D"
_graphicsinterface.org/Graphics_Interface/2020/Conference — Submitted to GI 2020_

### Official Review · AnonReviewer3 · 2020-01-02
**Unclear contribution, questionable evaluation**

**Confidence:** 4
**Rating:** 3

**Review:**

The paper's exposition is somewhat confuse. It is not easy to figure out what the paper actually aims to ultimately solve or show. Nowhere does it state unambiguously what it aims to achieve, what problem is to be solved or what question is to be answered, and in what sense the state of the art is supposed to be advanced. At some points in the paper (such as "In this paper we considered the set of spheres as shape descriptors") it seems that the paper's main proposal is to use the packed spheres (their radius and distance sequences?) as shape descriptor. If this was the key intent of the paper, comparisons with some of the many previous 3D shape descriptors would be necessary (which the paper does not deliver) -- and the entire SRS-based motivation and context would make no obvious sense.

In terms of contribution, it appears that the paper proposes no novel algorithm. The greedy ('largest sphere first') sphere packing algorithm listed on page 5 was apparently described in other work by the author(s) (Anonymous 2019), and also mentioned in various previous papers on sphere packing problems, such as [2] or [1]. The paper's contribution therefore boils down to an empirical evaluation - of this sphere packing approach's rotation invariance (in a discrete voxelized setting).

This evaluation provides only limited insight. This is due to two reasons:
1) It is clear already from the very definition of the procedure that the result can be highly unstable with respect to tiny perturbations of the input (as, e.g., induced by the rediscretization when rotating).
2) The analysis performed is of very questionable quality: the paper does not state in detail what the analysis is supposed to evaluate, and does not justify the choice of measures reported and compared in the context of SRS. Concretely, what the paper evaluates is the amount of change in (the ordered list of) sphere radii and in (the ordered list of) distances of consecutive sphere's centers. It remains unclear why invariance of these two measures is of relevance - whether in the SRS context or otherwise. After all, two geometrically identical (just differently ordered) sphere packings can differ largely in these measures, while two geometrically very different packings can be indistinguishable from these measures alone. The paper's claim that this is a "useful" "approximation" is not underpinned. The choice of distance ratios instead of distances is not justified either. Furthermore, the paper only considers whether identical shapes (up to rotation) have similar values, not whether different shapes have dissimilar values; this is not a reasonable way to evaluate a shape descriptor (a trivial 0-descriptor would be considered perfect under the paper's methodology).

There are various language issues. Most parts remain understandable, but several sentences are hard to make sense of. At several points it is unclear whether there is a language issue or there is a formally incorrect statement. One example is the sentence "There are three major techniques to prove the rotation invariance"; what follows is not a list of proof techniques. It did not become clear how the following landmarking/alignment discussion is related to the paper's content. There are further formally imprecise or incorrent statements, such as "our algorithm ... is defined as a set of ... spheres", "sphere centers ... respresent a spatial template as a graph", "intersection of the sphere's centers", or "a measure called epsilon-rotation invariant".

"Experimental results demonstrate the effectiveness and efficiency of the proposed method": neither the effectiveness nor the efficiency relative to the state of the art is demonstrated in the paper. It furthermore did not become clear what precisely is actually meant by "the proposed method" in the context of this paper. "... are actually within epsilon value criteria": this appears to be a trivial statement as epsilon is nowhere specified; it appears that any outcome could be considered to meet this criterion; there is always some value such that all observed errors are smaller. "high probability that the 3D volumes are similar": the paper does not establish this.

The difference in radii before and after rotation is put in relation to the "risk of developing recurrent disease"; it remained unclear what basis this statement is made on.

The title is inadequate because the paper does not describe a novel spheres packing algorithm - and the algorithm it describes is not epsilon-rotation invariant in any reasonable sense of the term (and a formal definition is not given in the paper).

The paper motivates the sphere packing problem that it considers using SRS. The description of that technique and the problem's relation to it are, however, not clear; a reader not familiar with the detailed principle of SRS will not understand the description in the paper. Previous work in the field, such as [1] or [2] does a much better job at concisely conveying the relation and relevance.

The conclusion starts with "Our novel medical visualization technique". It remains unclear how that is related to the shape descriptor focused paper content.

Comparison to previous approaches to the sphere packing problem, such as [1], [2], or previous work on shape descriptors, is missing.

The detailed (and very toolset specific) description of the implementation is unnecessarily verbose. E.g., there is no need to spell out standard 3D rotation matrices, etc.

The discussion of related work is insufficient. It focuses on shape descriptors but almost entirely ignores SRS and previous work on sphere packing and related problems.

There is no good reason to mention "Slicer3D" in the title, considering that the described method and evaluation are not Slicer3D-specific in any way.

The paper does not discuss the fact that the rotation variance comes from discretization artifacts. It does not explain why invariance (in particular invariance of radii and ratios of center distances) to rotation-and-rediscretization induced shape perturbations is relevant in any context.



[1]: "Packing of Unequal Spheres and Automated Radiosurgical Treatment Planning" by Wang (1999).
[2]: "Use of Shape for Automated, Optimized 3D Radiosurgical Treatment Planning" by Bourland and Wu (1996).

---

### Official Review · AnonReviewer1 · 2020-01-04
**Marginal Contribution, Unclear Problem Statement**

**Confidence:** 5
**Rating:** 2

**Review:**

The problem statement of this paper is very vague. Apparently, authors have already published a greedy sphere packing and they try to make it rotation invariant or experiment its properties under an arbitrary 3D rotation. Reading the paper, I could not figure out what are the challenges of this problem. Why if we have a set of spheres, we cannot simply rotate all of them against a point or axis and preserve the desired properties such as distance between the sphere centers (apparently this was the key in defining the feature). Is it due to re-voxelization? Why sphere packing at the first place? Why not simply using voxels? ... The paper has not been motivated well.

Aside form the lack of clarity of the intention of the paper, it suffers from presentation issues. Teaser has no informative caption and it is confusing to see the first Figure with graphs that are not fully understandable and in the text there is no reference to this figure. The same for many other figures in the paper: no reference in the text, no informative caption (e.g., Figure 4, 5, 6, 8, etc).

Introduction is a mix of related work and motivation and the paper suffers from the lack of having an actual related work and also a good motivation. There are many papers related to shape descriptors and also sphere packing and they have not been cited in the paper.

Unnecessary/obvious information is given. Providing rotation along different axis is not necessary. These are common knowledge in the Graphics community and they are not needed to be repeated in the paper. Instead the algorithm (which does not have a number in the paper) needs more explanation and a better presentation.

Aside from many problems about presentation and also technical issues, the contribution of the paper is unknown and marginal and I do not believe that it advances Graphics state of the art. The paper is clearly below the GI acceptance bar.

---

### Official Review · AnonReviewer2 · 2020-01-08
**Unstable method, lack of structure and comparisons**

**Confidence:** 4
**Rating:** 2

**Review:**

The paper is a little hard to follow due to poor grammar and wording as well as a general lack of structure and focus. It is not really clear what the authors want to achieve and why it is relevant. As far as I understand,
the idea of the paper is to use the output of a greedy sphere packing algorithm for 3d volumes to define a rotation invariant shape descriptor. To this end a voxelization of the object is used and spheres centered at voxels are greedily added such that they are inside the object, have maximal radius and don't intersect with previous spheres. The distances of consecutive spheres are then used to compare shapes.

I see several problems with this approach. I agree that for an arbitrarily fine discretization the method could identify different rotated versions of the same object. Since the discretization level will be limited, it can always happen, that a rotation of the object will lead to a completely different relative sphere placement, affecting all subsequent spheres arbitrarily. So there will be no guarantees for finding pairs of rotated shapes. This instability can also lead to vastly different signatures for slightly deformed objects, which is highly undesirable for a shape signature. Just checking 30 datasets does not convince me that this is not an issue.

Another problem is the complete lack of comparison with respect to competing methods. At least a comparison to a very basic algorithm, like alignment using PCA or ICP, is necessary here. The implementation section on the other hand is far to detailed and presents some details that are almost trivial in my opinion and do not contribute to the understanding of the algorithm.

Figures 9-11: It is not a good idea to put the patient ids on the x-axis and plotting results as a piecewise linear graph. The shape of this graph does not convey any information. A histogram plot should be used here.

In conclusion, I do not think that the presented approach is a useful shape descriptor. Unfortunately the evaluation could not convince me of the opposite. Therefore the paper should not be accepted.

---

### Meta-Review · Area_Chair1 · 2020-01-09

**Recommendation:** Reject
**Confidence:** 5

**Metareview:**

The reviewers agree that the problem statement is unclear; the paper lacks structure, focus, and motivation; the contribution is marginal; the discussion of and comparison with previous work is insufficient. The reviewers were not convinced that, considering the state of the art, the proposed use of voxel-based sphere packing to define a shape descriptor is a reasonable or fruitful idea, given its inherent instability.

---

### Decision · Program_Chairs · 2020-01-11

Reject